# Induction of competent cells for *Agrobacterium tumefaciens*-mediated stable transformation of common bean (*Phaseolus vulgaris* L.)

Guo-qing Song[1]*, Xue Han[1], Andrew T. Wiersma[2], Xiaojuan Zong[1], Halima E. Awale[2], James D. Kelly[2]

1 Plant Biotechnology Resource & Outreach Center, Department of Horticulture, Michigan State University, East Lansing, Michigan, United Sates of America, 2 Department of Plant, Soil, and Microbial Sciences, Michigan State University, East Lansing, Michigan, United States of America

* songg@msu.edu

**Data Availability Statement:** All relevant data are within the paper and its Supporting Information files.

## Abstract

Stable transformation of common bean (*Phaseolus vulgaris* L.) has been successful, to date, only using biolistic-mediated transformation and shoot regeneration from meristem-containing embryo axes. In this study, using precultured embryo axes, and optimal co-cultivation conditions resulted in a successful transformation of the common bean cultivar Olathe using *Agrobacterium tumefaciens* strain EHA105. Plant regeneration through somatic embryogenesis was attained through the preculture of embryo axes for 12 weeks using induced competent cells for *A. tumefaciens*-mediated gene delivery. Using *A. tumefaciens* at a low optical density (OD) of 0.1 at a wavelength of 600 nm for infection and 4-day co-cultivation, compared to $OD_{600}$ of 0.5, increased the survival rate of the inoculated explants from 23% to 45%. Selection using 0.5 mg L$^{-1}$ glufosinate (GS) was effective to identify transformed cells when the *bialaphos resistance* (*bar*) gene under the constitutive 35S promoter was used as a selectable marker. After an 18-week selection period, 1.5% -2.5% inoculated explants, in three experiments with a total of 600 explants, produced GS-resistant plants through somatic embryogenesis. The expression of *bar* was confirmed in first- and second-generation seedlings of the two lines through reverse polymerase chain reaction. Presence of the *bar* gene was verified through genome sequencing of two selected transgenic lines. The induction of regenerable, competent cells is key for the successful transformation, and the protocols described may be useful for future transformation of additional *Phaseolus* germplasm.

## Introduction

Common bean (*Phaseolus vulgaris* L.) is a member of the legume family Fabaceae. It is one of the most important grain legumes for direct consumption in human diets due to its unique nutritional profile and superior health benefits [1–4]. In response to increasing demands for

**Funding:** The authors received no specific funding for this study.

**Competing interests:** The authors declare that there is no competing interest.

food caused by the growing population, a fundamental goal of common bean breeding has been to develop high-yielding cultivars with desirable consumer traits [5]. A summary of the broad breeding objectives in common bean improvement was recently reviewed by Assefa [6]. Abundant genetic diversity, easy germplasm accessibility, and application of marker assisted breeding technologies have supported conventional bean breeding efforts [3, 7, 8]. In addition, advances in whole genome sequencing and transcriptome analysis have paved a way for genomics-enabled bean breeding through either conventional approaches or genetic engineering [9–11].

Genetic engineering is a powerful tool to introduce genes from sources that are inaccessible through sexual hybridization [12–14]. Thus, substantial efforts have been made to develop reliable transformation methods for engineering common beans with various traits [13, 15–21]. To date, no genetically engineered (GE) common bean has been commercialized, despite of the regulatory approval for the transgenic "Embrapa 5.1" common bean for golden mosaic virus (BGMV) resistance in Brazil in 2011 [22–24]. An efficient and reproducible transformation system for the production of stable transgenic common bean plants is still lacking. The primary limitation has been recalcitrance of common bean genotypes to *in vitro* regeneration from non-meristem containing tissues [16, 25, 26].

Stable transformation of common bean at low frequencies ($< 1\%$) has been achieved using particle bombardment-mediated transformation of meristematic tissues for different goals [11, 27, 28]. *Agrobacterium tumefaciens*-mediated transformation is desirable for common bean transformation because of its accessibility and tendency to produce low- or single-copy insertion(s) of the transgene [18]. To date, *A. tumefaciens*-mediated transformation of common bean has been studied extensively but has been met with limited success [16, 25]. We have been working on common bean regeneration and transformation since 2008. The main focus of this study was to develop a protocol for stable transformation of common bean using *A. tumefaciens*. The primary issue that arise from transformation of meristem-containing embryo axes is chimeric tissues; and it is difficult to produce a stable transgenic plant from a single transformed cell in $T_0$ generation [29–31]. In contrast, in this study, preculture of embryo axes induced competent cells for *A. tumefaciens*-mediated gene delivery and for nonchimeric regeneration through somatic embryogenesis. This is the key for production of stable transgenic pinto bean using *A. tumefaciens* in this study. The protocol described here has a potential application for transformation of other large seeded legumes, for which regenerable cells are limited to those from meristem-containing embryo axes (Fig 1).

## Materials and methods

### Plant materials and explant preparation

'Olathe' pinto bean (*Phaseolus vulgaris* L.) was used in this study due to its high efficiency in shoot production from mature embryo axes in our previous test (unpublished data). Between 200 and 300 mature, dry seeds were surface-sterilized with 2% sodium hypochlorite with gently shaking for 10 min in a 500 ml PYREX® media bottle (Corning, NY, USA) and followed by four rinses with sterile distilled water. The seeds were transferred to 4 or 5 Petri dishes (100 x15 mm) and each had one layer of seeds. Sterile distilled water, 30 ml/dish, was added to soak the seeds overnight at room temperature. All imbibed seeds were transferred to a sterile 500 ml PYREX® media bottle, where the seeds were washed three times with sterile distilled water prior to transferring to Petri dishes (100 x15 mm) for embryo axes extraction.

The seed coat was removed, and two types of explants were initially prepared after removing one cotyledon along the hilum using a sterile scalpel blade. A half-seed explant was prepared by removing the seed coat, radicle, or one cotyledon. Embryo axes explants were

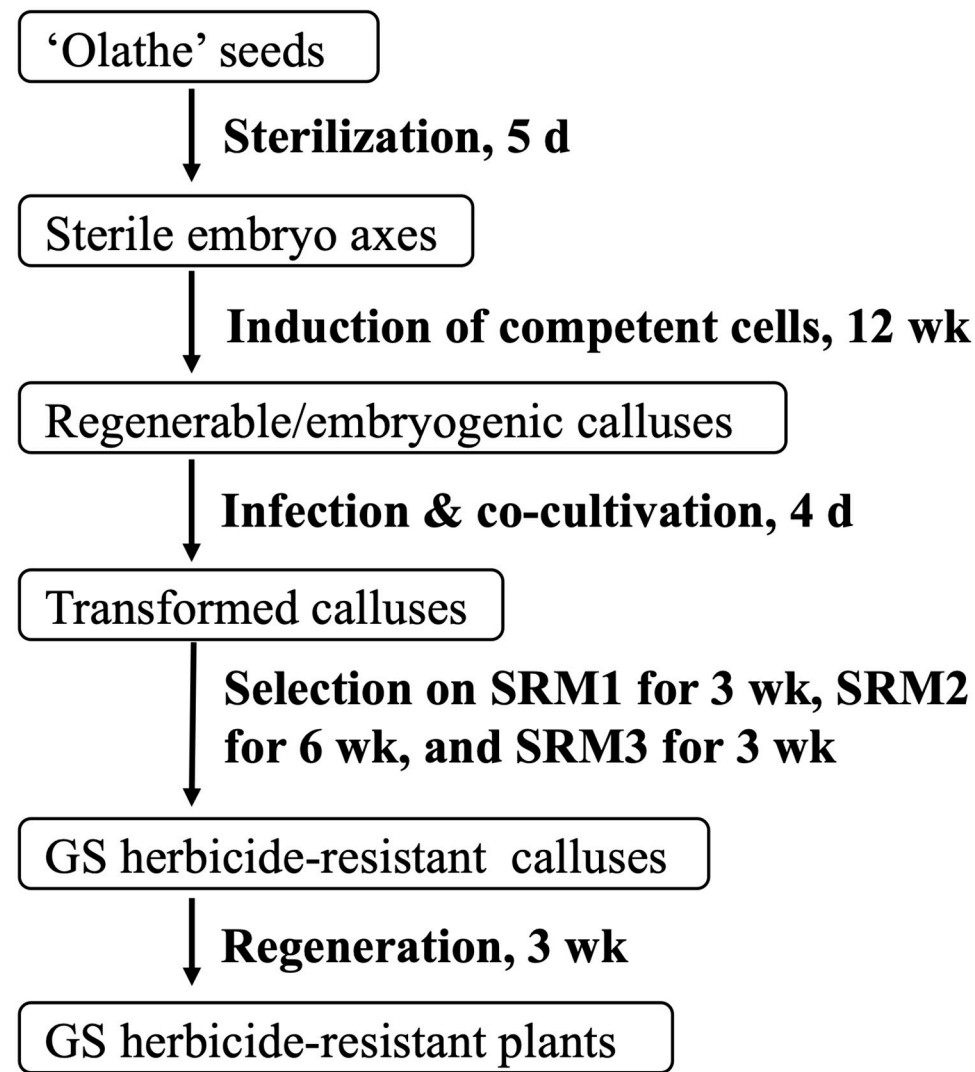

**Fig 1. Schematic representation of *A. tumefaciens*-mediated transformation of 'Olathe' pinto bean using competent cells induced from embryo axes.**

obtained by cutting off all cotyledons, radicles, and leaflets. To avoid potential cross-contamination, one set of sterile cutting tools (*i.e.*, a scalpel, a cutting board, and a pair of forceps) were used to excise no more than 5 seeds. Both half-seed explants and embryo axes were cultured for 5 days on Murashige and Skoog (1962) medium (MS) [32]. Sterile explants were used for transformation (Fig 1).

Preculture of embryo axes was conducted on 30 ml regeneration medium [RM: MS + 44.4 μM 6-benzyl-aminopurine (BAP) + 2.27 μM thidiazuron (TDZ)] in each Petri dish (100 x15 mm). Subculture of the axes to fresh RM was performed every three weeks (Fig 1). All shoots from the axes were removed using a sterile scalpel blade prior to each subculture. After 12 weeks, each precultured embryo axes was cut into 4 to 8 explants for transformation. All *in vitro* materials were placed at 25 ˚C under a 16 h photoperiod of 30 μmol m$^{-2}$s$^{-1}$ unless indicated otherwise. Both MS and RM media contained MS salts and vitamins, 3% sucrose, pH was adjusted to 5.6, and solidified with 0.6% (w/v) agar. When acetosyringone, antibiotics, and

glufosinate (GS) were used, they were filter-sterilized through 0.22 μm Millex®-GV filters (Millipore Corporation, Billerica, MA, USA) and added to media cooled to 50–60 ˚C after autoclaving.

## Plasmid and *Agrobacterium* strain

The binary vector pDHB321.1 contains only the *bialaphos resistance* (*bar*) gene under the constitutive 35S promoter in its transfer DNA region. The vector was transferred into *A. tumefaciens* strain EHA105 [33]. Expression of the *bar* gene conferring to glufosinate herbicide-resistance has been demonstrated previously in transgenic celery and blueberry plants [34, 35].

To prepare *A. tumefaciens* culture, a single colony of EHA105:pDHB321.1 was cultured in 10 mL liquid yeast extraction broth (YEB) [36] containing 100 mg $L^{-1}$ kanamycin monosulfate at 28 ˚C with constant shaking for 48 h. Thirty μL of the culture were then inoculated into 15 mL of the same medium and grown to an $OD_{600}$ of 0.8–1.0. Before transformation, the culture was centrifuged at 2500 ×g for 5 min. The bacterial pellet was resuspended to an $OD_{600}$ of 0.1 or 0.5 in liquid RM containing 100 μM acetosyringone.

## Transformation experiments

Explants were transferred to a 50 ml Corning tube. EHA105:pDHB321.1 suspension in liquid RM containing 100 μM acetosyringone were added to the Corning tube and the explants were inoculated for 2 min at room temperature. After inoculation, the suspension cells were poured off and the infected explants were transferred onto two layers of sterile filter paper in a Petri dish, where the explants were blotted dry. The explants were then placed on one layer of sterile filter paper overlaid on 25 ml RM containing 100 μM acetosyringone in a Petri dish. Co-cultivation was carried out for 4 d at 25 ˚C in the dark (Fig 1).

After co-cultivation, all explants were transferred into a 50 ml Corning tube and rinsed five times, 1 min per time, in liquid RM. The sixth wash was conducted in liquid RM containing 500 mg $L^{-1}$ timentin (Tn) and 500 mg $L^{-1}$ cefotaxime (Cef) for 3 min. Three selection media were used in the whole selection process (Fig 1). The washed explants were blotted dry on sterile filter paper in a Petri dish and place on selection RM1 [SRM1: RM containing 0.1 mg $L^{-1}$ GS, 250 mg $L^{-1}$ Tn, and 250 mg $L^{-1}$ Cef] and cultured for 3 wk. After 3 wk selection on RM1, the explants were subcultured to selection RM2 [SRM2: RM containing 0.2 mg $L^{-1}$ GS, 250 mg $L^{-1}$ Tn, and 250 mg $L^{-1}$ Cef] and a subculture to fresh SRM2 was conducted after 3 wk. After a total of 6 wk selection on SRM2, the explants were transferred to selection RM3 (SRM3: RM containing 0.5 mg $L^{-1}$ GS, 250 mg $L^{-1}$ Tn, and 250 mg $L^{-1}$ Cef). For each of the first three subcultures described (*i.e.*, from SRM1 to SRM2, from SRM2 to SRM2, and from SRM2 to SRM3), all emerged shoots for any explants were removed. The fourth subculture was a direct transfer of the explants to fresh SRM3 without removing any emerged shoots. The fifth subculture was to transfer the shoots or plants to 30 ml selection MS containing 0.5 mg $L^{-1}$ GS, 250 mg $L^{-1}$ Tn, and 250 mg $L^{-1}$ Cef in each 40 mm × 110 mm (diameter × height) glass jar for rooting. Shoots/plants from different explants were labelled separately. All subcultures were performed at 3 wk intervals.

Rooted GS-resistant plants were washed using tap water to remove any agar attached to the roots, each individual plant was then planted in water-soaked Suremix Perlite planting medium (Michigan Grower Products Inc., Galesburg, MI) in a 4-inch plastic pot (8.9 cm width × 12.7 cm height). Each pot was enclosed in a zipped one-gallon zip bag to keep moisture for one week. The bags were open progressively in one week. The survived plants were repotted to clay pot (22 cm width × 19 cm height). The plants were grown at 23–25 ˚C under a

16 h photoperiod of 55 µmol m$^{-2}$s$^{-1}$ and were watered and fertilized as needed to harvest $T_1$ seeds.

Following the protocols described, four transformation experiments were conducted to evaluate the effect of explant types and *Agrobacterium* concentrations on stable transformation. In the first experiment, half-seed explants and embryo axes explants were inoculated with EHA105:pDHB321.1 suspension at OD$_{600}$ of 0.1 and 0.5 for each type of explants. In the rest three experiments, EHA105:pDHB321.1 suspension at OD$_{600}$ of 0.1 was used. In the second experiment, both embryo axes and precultured embryo axes were inoculated. In the third experiment, only precultured embryo axes were infected to validate the best approach observed in the second experiment. Finally, in the fourth experiment, all three types of explants were used to compare different approaches at the same time; meanwhile, uninoculated explants, 60–90 for each type, were used as control to test their regeneration under the selection conditions.

## Phenotyping of $T_1$ and $T_2$ plants

Three $T_1$ seeds of transgenic line #1 and four $T_1$ seeds of transgenic line #2, one for each clay pot (22 cm width × 19 cm height), were sown. $T_1$ plants were grown in a growth chamber from March to June 2018. The growth chamber was set to 14 h photoperiod with 26/22 ˚C day and night temperature, respectively. Nontransgenic plants were used as a control. Plant growth was documented by taking pictures of the plants periodically. Self-pollinated $T_2$ seeds from each $T_1$ plant were harvested separately. Six $T_2$ seeds from each of the seven $T_1$ plants were grown in the greenhouse from July to October 2018 to bulk seeds.

## Molecular analysis

Genomic DNA from various tissues, *i.e.*, $T_0$ leaf, $T_0$ root, and $T_1$ leaf, was isolated using a cetyl trimethyl ammonium bromide (CTAB) method. Non-transgenic leaf and root were used as controls. A pair of primers, 35S_F: 5'-TGA CGC ACA ATC CCA CTA TC-3' and Bar_R1: 5'-GAA GTC CAG CTG CCA GAA AC-3', was used in a regular polymerase chain reaction (PCR) analysis for amplifying a 400-base pair (bp) fragment.

Total RNA was isolated from leaves of $T_1$ plants using the RNeasy Mini Kit (Qiagen, Valencia, CA, USA). All RNA samples were treated by DNase. PCR analysis of the RNA samples using 35S_F and Bar_R1 primers were conducted to detect the potential DNA contamination. Nontransgenic leaf and root were used as controls. Reverse transcription of RNA to complementary DNA (cDNA) was performed using SuperScript II reverse transcriptase (Invitrogen, Carsbad, CA, USA). 25–50 ng cDNA per reaction was used for reverse transcription (RT) PCR reactions. A pair of primers to amplify the full length of a 552 bp fragment, Bar_F: 5'-ATG AGC CCA GAA CGA CGC C-3' and Bar_R: 5'-TCA GAT CTC GGT GAC GG-3', was used for RT-PCR analysis. PCR conditions for both regular PCR and RT-PCR consisted of an initial denaturation at 94 ˚C for 2 min, followed by 30 cycles of 94 ˚C for 40 sec, 58 ˚C for 1.5 min, and 72 ˚C for 1.5 min, followed by a final extension at 72 ˚C for 10 min.

The DNA samples from one $T_0$ plant of the transgenic line 1, one $T_1$ plant of transgenic line 2, and one nontransgenic plant were purified for sequencing using the DNeasy Plant Mini Kit (Qiagen, Valencia, CA). Three samples were sequenced for 150-bp pair end reads (Illumina HiSeq 4000) with about 45-fold common bean genome coverage at the Research Technology Support Facility at Michigan State University [10]. ABySS/2.1.5 (k = 96) were used to assemble genome sequences using the resources at the High Performance Computing Center at Michigan State University [37]. The full length of the 35S-*bar* plus the nos terminator sequence was

used to search three assembled genome databases for the *bar* insertion using nucleotide Basic Local Alignment Search Tool (BLAST) [38].

## Results

### Induction of competent cells for common bean regeneration

Sterile embryo axes of 'Olathe' pinto bean produced an average of 3.6 shoots per explant after 4-week culture, whereas no regeneration occurred from leaf explants (Fig 2A and 2B). Interestingly, continuous removal of any shoots from embryo axes stimulated production of more callus-like green buds from all explants in 6–8 weeks (Fig 2C and 2D). These tiny buds can continue to proliferate during the subcultures on RM, thus providing an efficient approach to produce more buds. More importantly, these green buds were regenerable after they were transferred to MS medium. The tiny regenerable callus-like tissues obtained through preculturing embryo axes are desirable explants for transformation due to the increased number of competent cells for regeneration (Fig 2C and 2D). Each precultured embryo axes provided 4–8

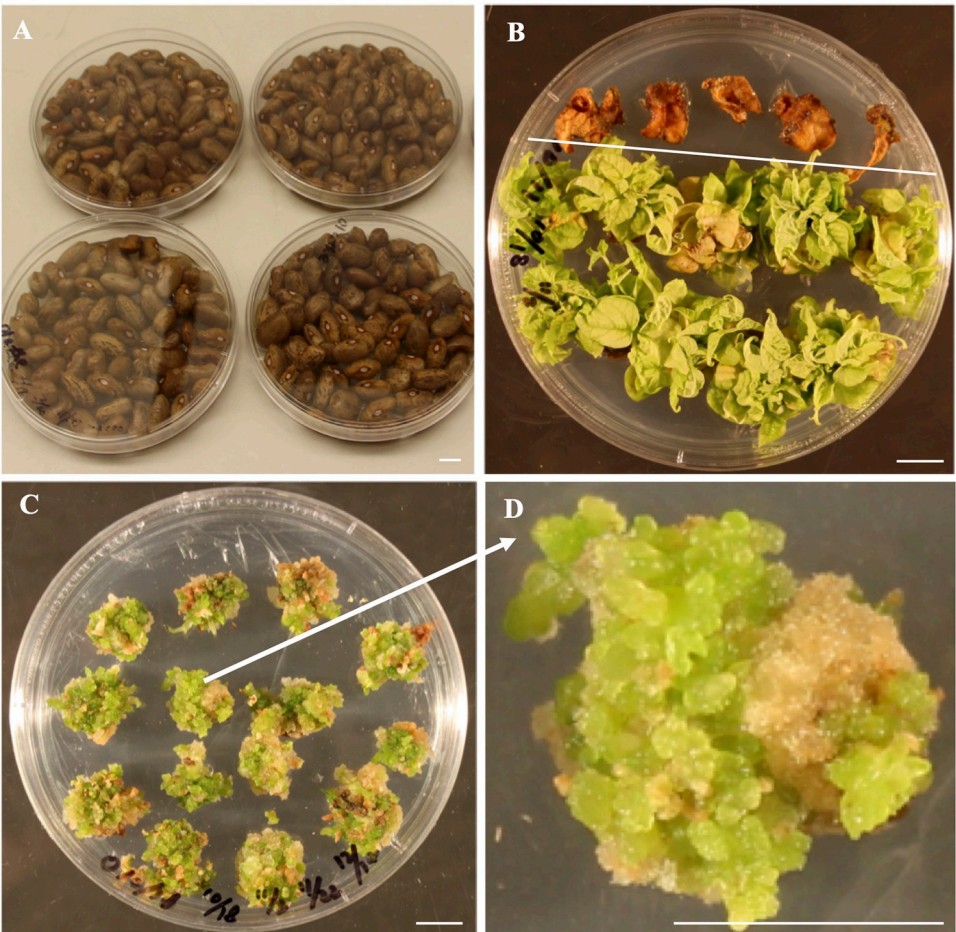

**Fig 2. Induction of competent cells from embryo axes for efficient regeneration of 'Olathe' pinto bean.** (A) Surface sterilized and soaked seeds for explant preparation. (B) Shoot production from embryo axes after 4-week culture on RM1, on which leaf explants did not survive. *A. tumefaciens*-mediated transformation of 'Olathe' pinto bean. (C, D) Embryo axes after 12-week preculture on RM. All shoots were removed during the subcultures in order to promote callus and bud formation. *Bars* = 1 cm.

regenerable callus-like explants for stable transformation; meanwhile, these precultured embryo axes can be subcultured and used for transformation even after 12 weeks.

## Low *Agrobacterium* concentration increases explants survival rate after infection and co-cultivation

In experiment 1, we tested both half-seed explants and embryo axes explants, two commonly used explants for legume, to evaluate EHA105:pDHB321.1 cell suspension at $OD_{600}$ of 0.1 and 0.5 for infection without agroinfiltration. For each of the two explant types, the $OD_{600}$ of 0.1 resulted in doubling explants survival rates compared to the $OD_{600}$ of 0.5 after 6-week selection (Table 1). The result suggests that a lower *Agrobacterium* concentration of $OD_{600}$ of 0.1 can reduce explant death and is preferable to $OD_{600}$ of 0.5 for infection. Thus, EHA105:pDHB321.1 suspension at $OD_{600}$ of 0.1 was chose to use for stable transformation in the rest of the experiments. No GS-resistant whole plants were generated in the experiment 1 (Table 1).

## Selection and regeneration of transgenic plants

Four-day co-cultivation after the infection using EHA105:pDHB321.1 suspension at $OD_{600}$ resulted an effective control of over-growth of the EHA105:pDHB321.1 during cocultivation and selection, respectively, for all of three explant types (Fig 3A and 3B). A gradual increase of

**Table 1. Summary of three transformation experiments of common bean 'Olathe'.**

|  | Explants | Infection | Number of inoculated explants | Number (percentage) of explants survived after 6-week of selection | Number of GS resistant shoots after 12-week selection | Number of GS resistant plants after 18-week selection | Transformation frequency |
|---|---|---|---|---|---|---|---|
| Experiment 1 |  |  |  |  |  |  |  |
|  | Half-seed explants | $OD_{600}$ of 0.1 | 65 | 26 (40.0%) | 1 | 0 | 0 |
|  |  | $OD_{600}$ of 0.5 | 63 | 12 (19.0%) | 0 | 0 | 0 |
|  | Embryo axes | $OD_{600}$ of 0.1 | 92 | 43 (46.7%) | 3 | 1 | 1.1% |
|  |  | $OD_{600}$ of 0.5 | 87 | 20 (23.0%) | 1 | 0 | 0 |
| Experiment 2 |  |  |  |  |  |  |  |
|  | Embryo axes | $OD_{600}$ of 0.1 | 200 | 84 (42.0%) | 8 | 1 | 0.5% |
|  | Pre-cultured embryo axes | $OD_{600}$ of 0.1 | 200 | 102 (51%) | 6 | 4 | 2% |
| Experiment 3 |  |  |  |  |  |  |  |
|  | Pre-cultured embryo axes | $OD_{600}$ of 0.1 | 200 | 90 (45%) | 4 | 3 | 1.5% |
| Experiment 4 |  |  |  |  |  |  |  |
|  | Half-seed explants | $OD_{600}$ of 0.1 | 100 | 33 (33.0%) | 2 | 0 | 0 |
|  | Embryo axes | $OD_{600}$ of 0.1 | 200 | 89 (44.5%) | 7 | 1 | 0.5% |
|  | Pre-cultured embryo axes | $OD_{600}$ of 0.1 | 200 | 112 (56.0%) | 8 | 5 | 2.5% |

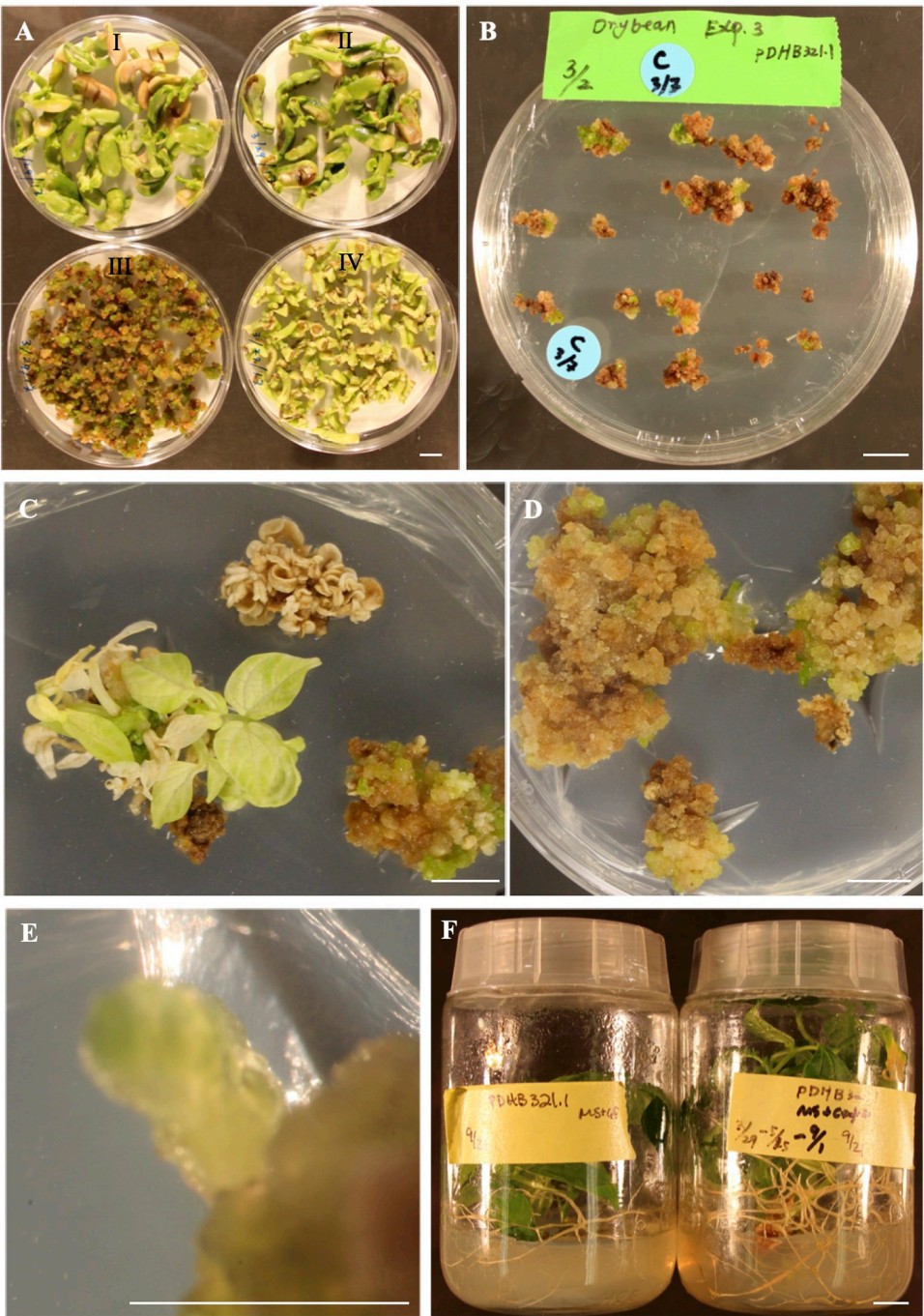

**Fig 3. *A. tumefaciens*-mediated transformation of 'Olathe' pinto bean.** (A) Explants after 4-day co-cultivation in the dark at 25 ˚C. (I, II) Half-seeds explants. (III) Precultured embryo axes explants. (IV) Embryo axes explants. (B) Precultured embryo axes explants after 3-week selection on selection RM1. (C) Embryo axes explants after 6-week selection on selection RM3. (D, E) GS-resistant calli and green buds induced from the precultured embryo axes explants after 6-week selection on selection RM3. (F) Rooting of the GS-resistant plants produced from the precultured embryo axes explants. *Bars* = 1 cm.

GS from 0.1 to 0.5 mg/L was effective in selecting transformed cells; meanwhile, it prevented the elimination of transformed cells by a high GS content before enough GS-resistance was developed (Fig 3B–3D). All shoots induced from embryo axes and half-seed explants within one-month selection using 0.1 mg/L GS turned white completely or partially after they were selected on RM3 containing 0.5 mg/L GS (Fig 3C), suggesting these early formed shoots were either escapes or chimeric. The result is consistent with the previous report in which early formed shoots following the transformation of the *gus*A reporter were either escapes or chimeric due to the none or partial GUS-staining [16]. This is reasonable because there was a very low possibility of plant/shoot formation from a single transformed cell of common bean within 4-weeks. Even in the model plant tobacco (*Nicotiana tabacum*), transgenic shoots are usually produced after 6-week selection. Thus, removing all the shoots formed within the first 12-week of selection was conducted to minimize production of the chimeric transformants in this study. After a total of 18-weeks of selection, no GS-resistant plants were obtained from the 165 inoculated half-seed explants inoculated in experiments 1 and 4. One GS-resistant plant was produced for each of the three experiments (Table 1).

In contrast to embryo axes and half-seed explants, precultured embryo axes explants did not produce many big shoots during the whole GS-selection process (Fig 3B). After 12-week selection, green bud- or somatic embryo-like regenerants were produced on selection RM3 (Fig 3D and 3E). These regenerants rooted on selection MS containing 0.5 mg L$^{-1}$ GS (Fig 3F). The overall selection, regeneration, and rooting process took 18 to 20 weeks and the frequencies of GS-resistant plants for three transformation frequencies were 2%, 1.5%, and 2.5%, because no rooted GS-resistant plants were produced from the uninoculated explants. The results of the three transformation experiments suggest that the protocol is reliable for a successful transformation although the transformation frequencies were not high. Apparently, the precultured embryo axes can produce more regenerable competent cells for transformation and regeneration through somatic embryogenesis (Fig 2C); consequently, these regenerable competent cells, unlike the other explants used in either this study or in literatures, are key for the successful transformations in this study (Table 1).

## Production of $T_1$ seeds and $T_2$ seeds

The main focus of this study was to develop transformation protocols, thus, effort with three experimental repeats (*i.e.*, experiments 2–4) was made to verify that successful transformations using precultured embryo axes explants were repeatable (Table 1). After the success in experiment 4, the first three rooted $T_0$ transgenic lines were transplanted to soil and easily survived. However, none of them set seeds due probably to overwatering. Subsequently, two additional transgenic lines were planted (Fig 4A and 4B), five and six $T_1$ seeds were obtained from transgenic line 1 and 2, respectively.

Three and four $T_1$ seeds for transgenic line 1 and 2, respectively, were planted in the greenhouse to bulk $T_2$ seeds (Fig 5A and 5B). All $T_1$ plants were grown normally and produced $T_2$ seeds (Fig 5C and 5D).

## Molecular analysis of $T_0$ and $T_1$ plants

PCR analysis was conducted using DNA samples isolated from both leaf and root tissues of all 15 $T_0$ transgenic plants produced on selection MS containing 0.5 mg L$^{-1}$ GS after 18 to 20 weeks across all four experiments. Thirty DNA samples from the 15 $T_0$ transgenic plants obtained in four experiments were PCR positive for the presence *bar* gene, whereas all eight DNA samples of nontransgenic leaf and root were PCR-negative for the *bar* gene. The result indicated that the GS-resistant plants contained transgenic cells in both leaf and root tissues.

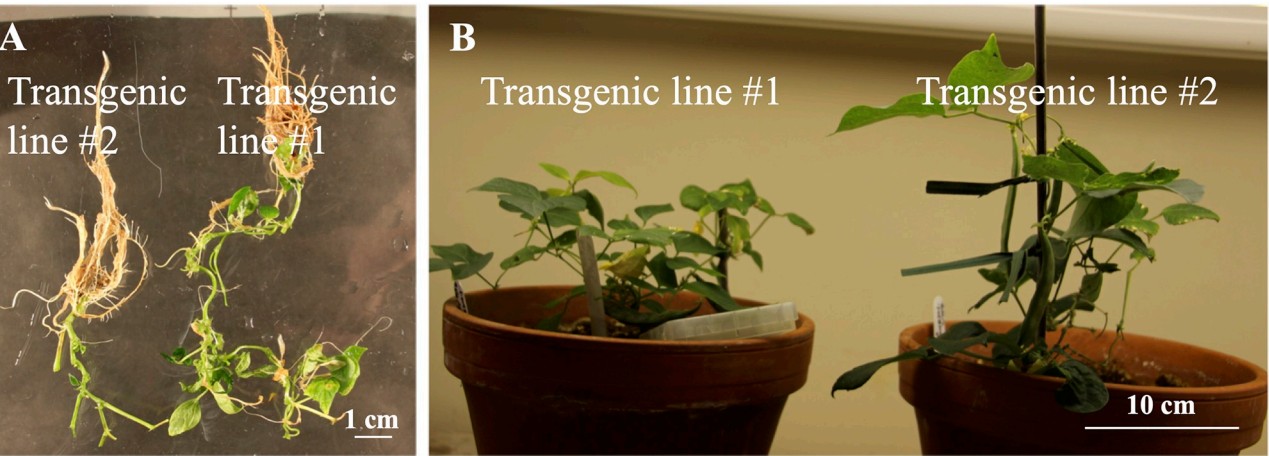

**Fig 4. Rooting of T₀ 'Olathe' pinto bean and T₁ seed production.** (A) Rooting of GS-resistant plants on selection MS. (B) Production of T₁ seeds at 23–25 °C under a 16 h photoperiod of 55 μmol m⁻²s⁻¹.

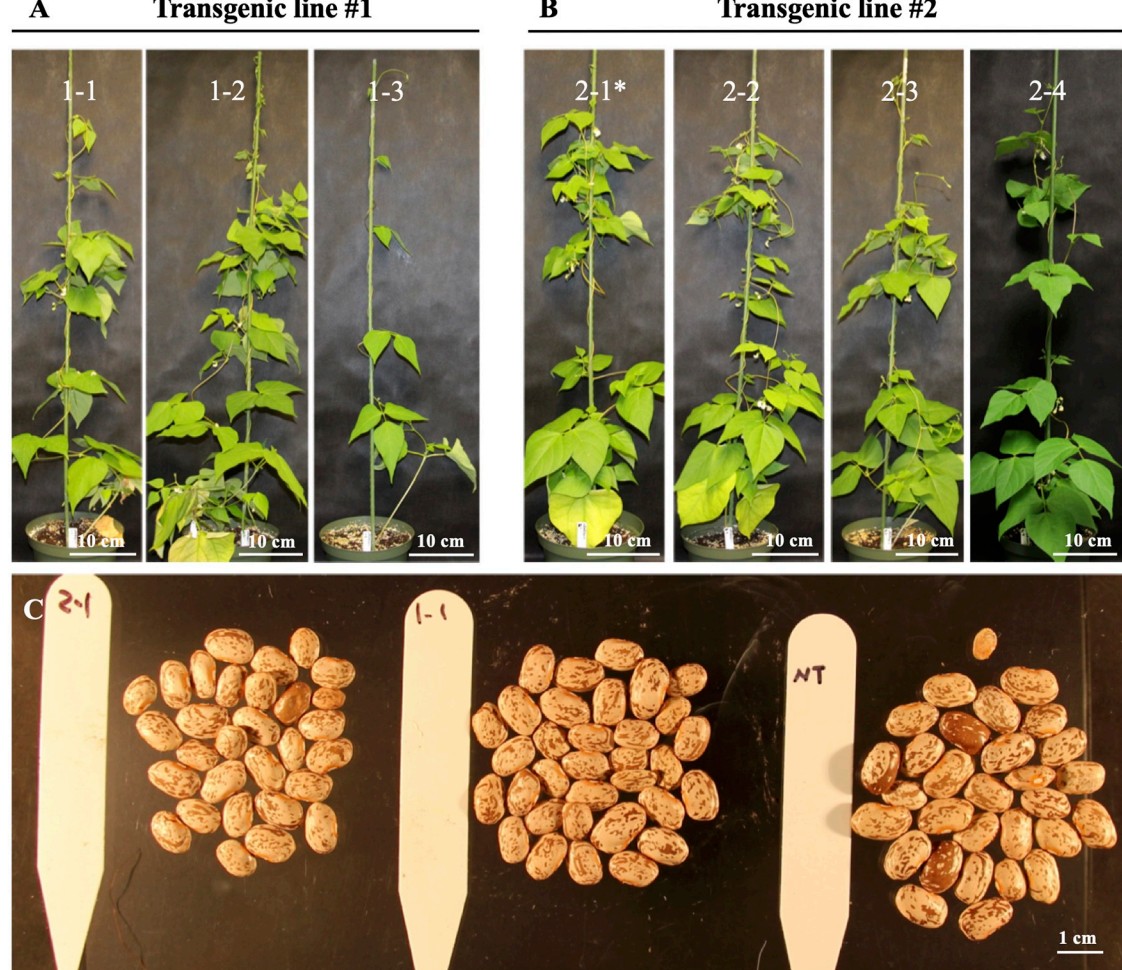

**Fig 5. T₁ plants and T₂ seeds.** (A, B) T₁ plants of two transgenic lines of 'Olathe' pinto bean. Except 2–1*, all the other plants are RT-PCR-positive for the *bar* gene. (C) Seeds from transgenic 1–1 and nontransgenic (NT) 2–1 and NT plants.

Based on the PCR results, the transformation frequencies in three experiments using embryo axes as explants were 1.1%, 0.5%, and 0.5%; and the transformation frequencies using precultured embryo axes were 2%, 1.5%, and 2.5%. Apparently, the precultured embryo axes led to higher transformation frequencies than the embryo axes likely due to the increased number of competent cells for transformation and regeneration in each explant.

Whether or not chimeric plants are produced, an effective transformation method for seed crops must result in the development of transgenic $T_1$ seeds, from which homozygous progeny can be identified and advanced. In this study, a total of seven $T_1$ plants, 3 and 4 for each of the two transgenic lines, were analyzed through PCR using the 35S_F and the Bar_R1 primers. Presence of the 400-bp fragment of the *bar* gene was observed in all three plants derived from transgenic line #1 and 3 out of 4 plants derived from transgenic line #2; in contrast, no fragment was detected in nontransgenic controls. The results suggested that the *bar* gene was present in six PCR-positive $T_1$ plants. Similarly, expression of the *bar* gene was verified in all six PCR-positive $T_1$ plants through RT-PCR using genomic DNA-free cDNA and the Bar_F and Bar_R primers (Fig 6). The PCR and RT-PCR results were consistent for the $T_1$ plants. The repeated success of utilizing precultured embryo axes as explants demonstrates the reliability of this transformation protocol for the pinto bean variety Olathe.

Whole genome sequencing was conducted using DNA samples from a non-transgenic plant and two transgenic lines, including one $T_0$ sample of transgenic line 1 and one $T_1$ sample from transgenic line #2. In the assembled genome sequences for both transgenic lines, the contigs containing the 35S-bar gene were present; in contrast, the 35S-bar gene was not detected in non-transgenic plant. In the $T_1$ transgenic plant, two contigs included a total of 91% of the sequence of the *bar* gene (Fig 7A and 7B). In the $T_0$ transgenic plant of the transgenic line #1, 500 contigs each contained 123–278 bp sequences aligned to the regions dispersed across the whole *bar* gene (S1 Table); surprisingly, some of these contigs had sequence overlaps but were

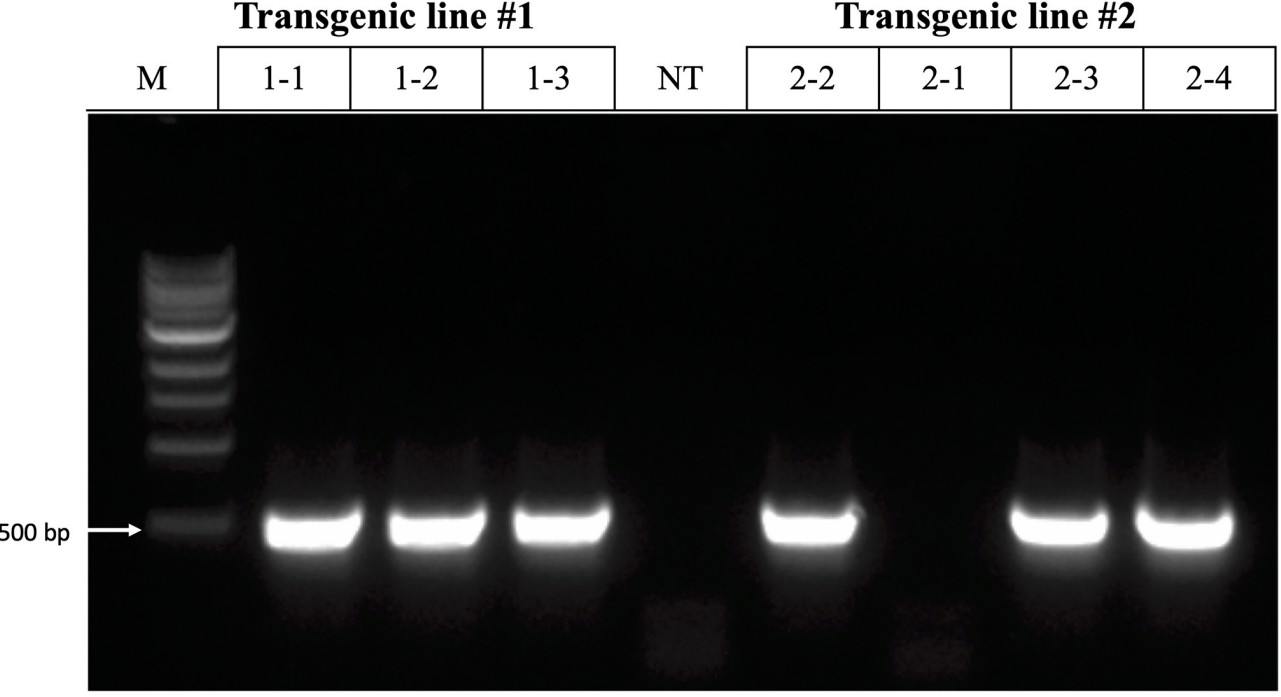

**Fig 6. RT-PCR analysis of *bar* expression in leaves of $T_1$ transgenic and nontransgenic (NT) plants of 'Olathe' pinto bean.** M: size ladder.

not assembled into longer contigs, which may be a limitation of the *de novo* sequence assembly approach. Based on the detected transgene contigs in the assembled genomes, it was likely that the transgenic line #2 had one-copy insertion and transgenic line #1 had multiple copies, although the insertion position(s) could not be identified. When pRiA4 sequence in the backbone region of pDHB32.1 was used to BLAST against the assembled genomes (Fig 7A and 7C), the pRiA4 was detected in both the non-transgenic and the $T_0$ transgenic plants; in contrast, a portion of the pRiA4 (3,169 out of 8,917 bp, e-value = 0) near the left border of the transfer DNA (T-DNA) was detected in the genome of the $T_1$ transgenic plant (Fig 7C). Additionally, sequences showed high similarities to the Kan$^R$ gene were not detected in the assembled genomes of any non-transgenic and transgenic lines. Apparently, an integration of a partial of the non-T-DNA pRiA4 occurred in the $T_1$ transgenic plant. If the detected transgene sequences were due to the residual of EHA105:pDHB321.1, we should have detected the Kan$^R$ gene sequences in the transgenic lines (Fig 7A). The results verify the stable transformation of the 35S-bar in the two transgenic lines for the first time using whole genome sequencing.

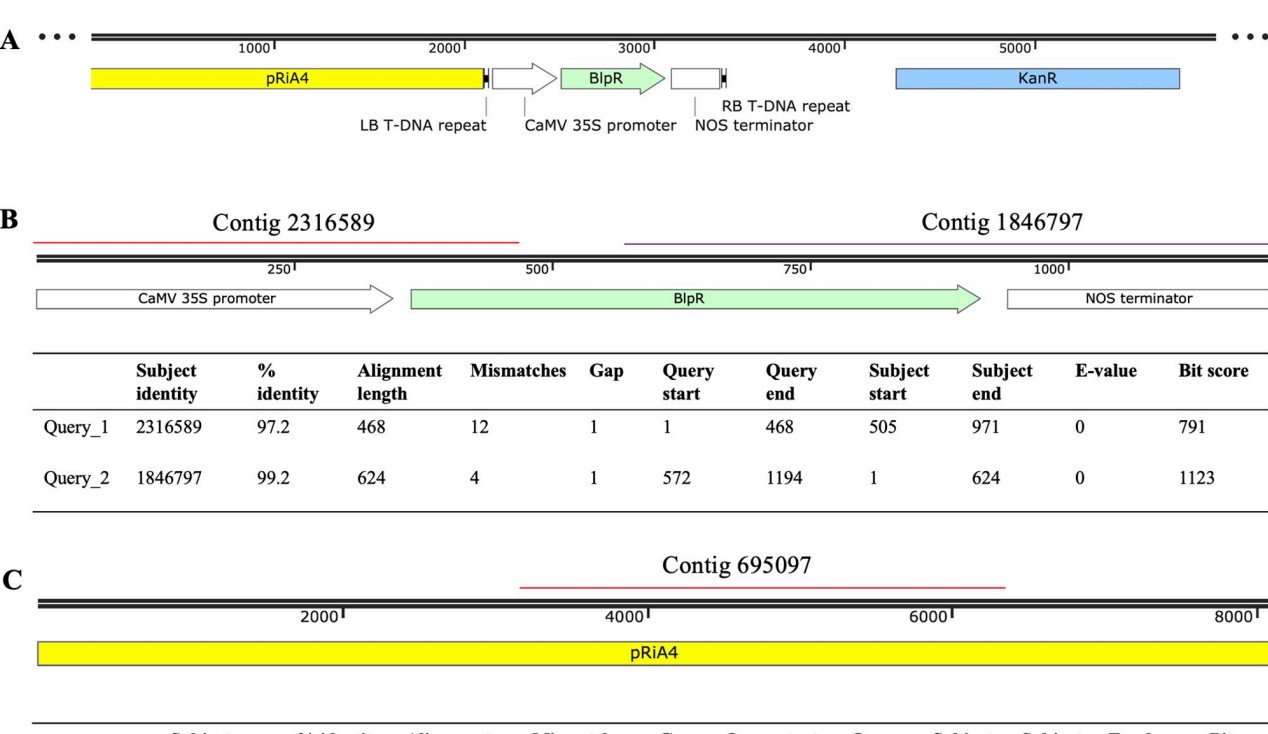

| | Subject identity | % identity | Alignment length | Mismatches | Gap | Query start | Query end | Subject start | Subject end | E-value | Bit score |
|---|---|---|---|---|---|---|---|---|---|---|---|
| Query_1 | 2316589 | 97.2 | 468 | 12 | 1 | 1 | 468 | 505 | 971 | 0 | 791 |
| Query_2 | 1846797 | 99.2 | 624 | 4 | 1 | 572 | 1194 | 1 | 624 | 0 | 1123 |

| | Subject identity | % identity | Alignment length | Mismatches | Gap | Query start | Query end | Subject start | Subject end | E-value | Bit score |
|---|---|---|---|---|---|---|---|---|---|---|---|
| Query_1 | 695097 | 84.6 | 3182 | 473 | 12 | 3684 | 6860 | 1 | 3169 | 0 | 3140 |
| Query_1 | 889580 | 81.0 | 1605 | 283 | 19 | 325 | 1920 | 1592 | 1 | 0 | 1254 |
| Query_1 | 630090 | 86.1 | 854 | 117 | 2 | 6793 | 7645 | 1 | 853 | 0 | 917 |
| Query_1 | 2530088 | 81.3 | 913 | 167 | 4 | 8007 | 8917 | 911 | 1 | 0 | 736 |
| Query_1 | 1245374 | 84.6 | 695 | 107 | 0 | 3031 | 3725 | 1 | 695 | 0 | 691 |
| Query_1 | 2519430 | 85.1 | 308 | 46 | 0 | 2363 | 2670 | 308 | 1 | 7.31E-83 | 315 |
| Query_1 | 692489 | 82.9 | 304 | 50 | 2 | 1958 | 2260 | 1 | 303 | 4.46E-70 | 272 |

**Fig 7. Stable integration of the transgenes confirmed by genome sequencing of a $T_1$ transgenic plant of the transgenic line #2.** (A) Schematic representation of the binary vector pDHB321.1. (B) Detection of the *bar* in the T-DNA region. (C) Detection of the pRiA4 sequence in the backbone region of the pDHB321.1.

## Discussions

### Meristem tissues are a reliable source of regenerable calli

Meristem-containing tissues remain the major regenerable explants for stable transformation of grain legumes, such as cotyledonary node or embryo axes for soybean (*Glycine max* (L.) Merr.) [39, 40], common bean [41], and cowpea (*Vigna unguiculata* L. Walp) [42]. In our efforts to induce shoots from the embryo axes of common bean cultivars, all of the five cultivars tested were able to produce multiple shoots [43], suggesting that the meristem tissues are amenable to shoot production either through proliferation of the original meristems or through new shoot organogenesis from the undifferentiated cells adjacent to the meristems. If new shoot organogenesis did occur, we hypothesized that a preculture of embryo axes on RM and continuous removal of any shoots from embryo axes would induce more cells that are amenable for both *A. tumefaciens*-mediated transgene(s) delivery and plant regeneration. This hypothesis was tested in a preliminary experiment using embryo axes of Olathe' pinto bean. We found that a 10-week culture of the embryo axes resulted in desirable cells similar to those in Fig 2C and 2D. The preliminary results laid the foundation of this research. Because recalcitrance of common bean genotypes to *in vitro* regeneration from non-meristem containing tissues is the primary limitation for stable transformation [16, 25, 26], the preculture method described for the Olathe' pinto bean may facilitate transformation of the other common bean cultivars. In fact, it has been reported that the EHA105-mediated transformation of primary and proliferative calli of common bean cv. CIAP7247F resulted in transgenic plants [20].

### *A. tumefaciens*-inoculation induces explants death

High concentrations of *A. tumefaciens*, agroinfiltration and a long cocultivation time enhanced transient expression of an intron-interrupted *gus*A in common bean cultivars [16]. However, in stable transformations, we previously found that the agroinfiltration using EHA105: pDHB321.1 suspension at $OD_{600}$ of 0.5 caused death of 70% of embryo axes explants in two weeks after co-cultivation under even a non-selection condition (unpublished data). Apparently, excessive *A. tumefaciens* infection through agroinfiltration could be responsible for explant death. Similarly, in the failure of our previous efforts in producing stable transgenic plants, the default inoculation conditions adapted from the other crops using EHA105 at $OD_{600}$ of 0.5–0.8 and 4-day co-cultivation led to unusual high death percentages (20%-50%) of the inoculated explants under effective controls of *A. tumefaciens* growth after co-cultivation. In these cases, we believed that the high EHA105 cell concentrations could be responsible for the death of the explants due to the lack of any other more reasonable explanation. Thus, the low EHA105 concentration at $OD_{600}$ of 0.1 was tested, and the results demonstrated that low EHA105 concentration contributed the successful transformations in this study (Table 1).

### Effectiveness of whole genome sequencing for stable transformation confirmation

When meristems were used as explants for common bean transformation, it is very common that chimeric transformants can be produced [43]. To verify the stable transgenic lines, Southern blot analysis of $T_1$ plants are often used. Alternatively, we found that whole genome sequencing was an effective approach to confirm stable transformations [44, 45]. In this study, although the assembled sequencing data did not allow us to determine the copy number as well as the insertion position, the detected T-DNA sequences verified the stable integration of the transgenes (Fig 7). To our knowledge, this is the first stable transgenic common bean

obtained through *A. tumefaciens*-mediated transformation and confirmed by whole genome sequencing.

## Conclusions

We obtained stable transgenic common bean through *A. tumefaciens*-mediated transformation. We found that the induction of competent cells prior to the infection and a low *Agrobacterium* concentration for infection and co-cultivation was important in our success in stable transformation. The stable transformation was verified through whole genome sequencing of the transgenic plants. This protocol has potential to be used for transformation of other important legume crops.

## Supporting information

**S1 Table. Identification of transgene inserts using 35S-bar as a query BLASTN against the assembled genome of the T$_0$ plant of transgenic line #1.**
(XLSX)

**S1 Fig.**
(TIF)

## Author Contributions

**Data curation:** Guo-qing Song, Xiaojuan Zong, James D. Kelly.

**Formal analysis:** Guo-qing Song.

**Investigation:** Guo-qing Song, Xue Han, Andrew T. Wiersma, Xiaojuan Zong, Halima E. Awale.

**Methodology:** Guo-qing Song, Xue Han.

**Project administration:** Guo-qing Song.

**Supervision:** Guo-qing Song.

**Writing – original draft:** Guo-qing Song.

**Writing – review & editing:** Guo-qing Song, Andrew T. Wiersma, Halima E. Awale, James D. Kelly.

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
