## [Decision Letter · Decision Letter 0]

9 Jan 2020

PONE-D-19-35407

Induction of competent cells for Agrobacterium tumefaciens-mediated stable transformation of common bean (Phaseolus vulgaris L.)

PLOS ONE

Dear Dr Song,

Thank you for submitting your manuscript to PLOS ONE. After careful consideration, we feel that it has merit but does not fully meet PLOS ONE’s publication criteria as it currently stands. Therefore, we invite you to submit a revised version of the manuscript that addresses the points raised during the review process.

We would appreciate receiving your revised manuscript by Feb 23 2020 11:59PM. To enhance the reproducibility of your results, we recommend that if applicable you deposit your laboratory protocols in protocols.io, where a protocol can be assigned its own identifier (DOI) such that it can be cited independently in the future. For instructions see: http://journals.plos.org/plosone/s/submission-guidelines#loc-laboratory-protocols

We look forward to receiving your revised manuscript.

Kind regards,

Jen-Tsung Chen, Ph.D.

Academic Editor

PLOS ONE

Journal Requirements:

'This research was primarily supported by Michigan State University through a Project GREEEN grant for infrastructure of the Plant Biotechnology Resource & Outreach Center.'

'The funders had no role in study design, data collection and analysis, decision to publish, or preparation of the manuscript.'

Additional Editor Comments (if provided):

Reviewers' comments:

Reviewer's Responses to Questions

**Comments to the Author**

1. Is the manuscript technically sound, and do the data support the conclusions?

Reviewer #1: Partly

Reviewer #2: Yes

2. Has the statistical analysis been performed appropriately and rigorously? 

Reviewer #1: Yes

Reviewer #2: N/A

3. Have the authors made all data underlying the findings in their manuscript fully available?

Reviewer #1: No

Reviewer #2: Yes

4. Is the manuscript presented in an intelligible fashion and written in standard English?

Reviewer #1: Yes

Reviewer #2: Yes

5. Review Comments to the Author

Reviewer #1: The paper should be deeply revised the structure of the paper according to Plos one instruction (“Results and discussions” should divided into results section and discussions section). In particular the Discussion section should be completely rewritten, should involve in “the transgenic “Embrapa 5.1” common bean for golden mosaic virus (BGMV) resistance”. In my opinion, the manuscript could not be accepted for publication in the present form.

1. The quality of photos (Fig.2-Fig.5, Fig. 7) is not high enough, and the label obscures the image, please replace.

2. The marker can't see at all in Fig.6, why? Please replace.

3. Line 37 “This is the first stable transgenic common bean”. If you have authentic proofs please provide that. Otherwise rewrite.

4. Check the small errors in the whole manuscript (for example: Line 50 “by”, Line 93 “, and”).

5. The first abbreviation shall provide the full name. For example, Line 173 “CTAB”.

6. Figure caption and citations should be referred to Plos one instruction.

Reviewer #2: The have developed an Agrobacterium-mediated transformation protocol for Phaseolus vulgaris. Although more efficient and less time-consuming approach for Phaseolus agrobacterial transformation already exist (Collado et al., 2015), this work is worth publishing, because it presents an alternative way that might serve for gene delivery to recalcitrant Phaseolus genotypes. In general, the manuscript is well written, concise and straightforward. In my opinion this manuscript can be accepted for publishing after the authors will revise/explain following issues:

1. Authors refer to picture 7A, B or C but nothing like these letters is presented in picture 7.

2. Line 293 - authors claim that T1 plants were subjected to both PCR and RT-PCR. I could not find any output of PCR analysis - it is not clear from text whether the PCR results where similar to RT-PCR or why do authors not present them in any table or image. Moreover, when presenting the RT-PCR results, the authors should show also the results of RT- (non-reverse transcribed) reaction to exclude the amplification from not properly removed genomic DNA (especially when the amplicon does not contain the intron).

3. Line 316 - authors claim that an integration of non-T-DNA fragment occured in T1 plants. I wonder how this can happen, despite the fact, that T1 generation plants were not subjected to Agrobacterium transformation.

6. PLOS authors have the option to publish the peer review history of their article (what does this mean?). If published, this will include your full peer review and any attached files.

Reviewer #1: No

Reviewer #2: Yes: Vojtech Hudzieczek

---

## [Author Response · Author response to Decision Letter 0]

30 Jan 2020

Included in the file "PONE-D-19-35407-review comments"

---

## [Decision Letter · Decision Letter 1]

19 Feb 2020

Induction of competent cells for Agrobacterium tumefaciens-mediated stable transformation of common bean (Phaseolus vulgaris L.)

PONE-D-19-35407R1

Dear Dr. Song,

We are pleased to inform you that your manuscript has been judged scientifically suitable for publication and will be formally accepted for publication once it complies with all outstanding technical requirements.

With kind regards,

Jen-Tsung Chen, Ph.D.

Academic Editor

PLOS ONE

Additional Editor Comments (optional):

Reviewers' comments:

Reviewer's Responses to Questions

**Comments to the Author**

1. If the authors have adequately addressed your comments raised in a previous round of review and you feel that this manuscript is now acceptable for publication, you may indicate that here to bypass the “Comments to the Author” section, enter your conflict of interest statement in the “Confidential to Editor” section, and submit your "Accept" recommendation.

Reviewer #1: All comments have been addressed

2. Is the manuscript technically sound, and do the data support the conclusions?

Reviewer #1: Yes

3. Has the statistical analysis been performed appropriately and rigorously? 

Reviewer #1: Yes

4. Have the authors made all data underlying the findings in their manuscript fully available?

Reviewer #1: Yes

5. Is the manuscript presented in an intelligible fashion and written in standard English?

Reviewer #1: Yes

6. Review Comments to the Author

Reviewer #1: (No Response)

7. PLOS authors have the option to publish the peer review history of their article (what does this mean?). If published, this will include your full peer review and any attached files.

Reviewer #1: No

---

## [Editor Report · Acceptance letter]

21 Feb 2020

PONE-D-19-35407R1 

Induction of competent cells for *Agrobacterium tumefaciens*-mediated stable transformation of common bean (*Phaseolus vulgaris* L.) 

Dear Dr. Song:

I am pleased to inform you that your manuscript has been deemed suitable for publication in PLOS ONE. Congratulations! Your manuscript is now with our production department. 

With kind regards,

on behalf of

Dr. Jen-Tsung Chen 

Academic Editor

PLOS ONE